# Development of a New Wearable Device for the Characterization of Hand Tremor

**DOI:** 10.3390/bioengineering10091025

**Published:** 2023-08-30

**Authors:** Basilio Vescio, Marida De Maria, Marianna Crasà, Rita Nisticò, Camilla Calomino, Federica Aracri, Aldo Quattrone, Andrea Quattrone

**Affiliations:** 1Biotecnomed S.C.aR.L., Viale Europa, 88100 Catanzaro, Italy; basilio.vescio@biotecnomed.it; 2Neuroscience Research Center, Department of Medical and Surgical Sciences, University “Magna Graecia”, Viale Europa, 88100 Catanzaro, Italy; m.demaria@unicz.it (M.D.M.); mariannacrasa@unicz.it (M.C.); r.nistico@unicz.it (R.N.); camilla.calomino@unicz.it (C.C.); federica.aracri@unicz.it (F.A.); quattrone@unicz.it (A.Q.); 3Institute of Neurology, Department of Medical and Surgical Sciences, University “Magna Graecia”, Viale Europa, 88100 Catanzaro, Italy

**Keywords:** tremor pattern, inertial signals, wearable device, machine learning, pattern prediction

## Abstract

Rest tremor (RT) is observed in subjects with Parkinson’s disease (PD) and Essential Tremor (ET). Electromyography (EMG) studies have shown that PD subjects exhibit alternating contractions of antagonistic muscles involved in tremors, while the contraction pattern of antagonistic muscles is synchronous in ET subjects. Therefore, the RT pattern can be used as a potential biomarker for differentiating PD from ET subjects. In this study, we developed a new wearable device and method for differentiating alternating from a synchronous RT pattern using inertial data. The novelty of our approach relies on the fact that the evaluation of synchronous or alternating tremor patterns using inertial sensors has never been described so far, and current approaches to evaluate the tremor patterns are based on surface EMG, which may be difficult to carry out for non-specialized operators. This new device, named “RT-Ring”, is based on a six-axis inertial measurement unit and a Bluetooth Low-Energy microprocessor, and can be worn on a finger of the tremulous hand. A mobile app guides the operator through the whole acquisition process of inertial data from the hand with RT, and the prediction of tremor patterns is performed on a remote server through machine learning (ML) models. We used two decision tree-based algorithms, XGBoost and Random Forest, which were trained on features extracted from inertial data and achieved a classification accuracy of 92% and 89%, respectively, in differentiating alternating from synchronous tremor segments in the validation set. Finally, the classification response (alternating or synchronous RT pattern) is shown to the operator on the mobile app within a few seconds. This study is the first to demonstrate that different electromyographic tremor patterns have their counterparts in terms of rhythmic movement features, thus making inertial data suitable for predicting the muscular contraction pattern of tremors.

## 1. Introduction

Tremor is the most common movement disorder and is defined as an involuntary, rhythmic, oscillatory movement of a body part, often involving the hands [1]. Differential diagnosis of tremor mainly comprises Parkinson’s disease (PD) and essential tremor (ET), which are the two most common causes of upper limb tremor [1,2,3,4].

Electrophysiological tremor analysis allows an objective and reproducible characterization of tremor features, which can help clinicians in the differential diagnosis [5,6]. Among the various tremor features, the contraction pattern of the antagonistic forearm muscles has shown the highest diagnostic value in previous studies, outperforming tremor amplitude or frequency, especially when tremor analysis was employed in patients with rest tremor (RT) [7]. 

The evaluation of rest tremor pattern is usually performed through visual inspection of tremor recordings by neurologists or neurophysiologists expert in tremor analysis [7,8]. In addition, a quantitative method also exists [9] to confirm the visual assessment of the pattern, based on the calculation of the tremor phase, which reflects the shift between contraction bursts of antagonist muscles; the pattern is synchronous (S) when bursts are “in phase” (low phase values) and alternating (A) when bursts are phase-shifted (high phase values). These methods are equally valid. However, both are based on the surface EMG tremor analysis, which requires an electromyograph and profound technical expertise (correct positioning of the electrodes, trace acquisition, visual interpretation of the tremor recordings, or quantitative phase analysis). For this reason, to date, the analysis of tremor patterns and phases has been performed mainly in centers that have access to technological and human resources. In Figure 1, example tremor EMG signals are shown, acquired by an electromyograph, from two different patients: the signal in Figure 1A refers to an alternating tremor pattern, while the signal in Figure 1B is an example of a synchronous tremor activation pattern.

In the last years, there has been an increasing demand for more accessible tools and techniques that could assist not only specialists but also general practitioners in screening for neurological disorders characterized by tremor symptoms. In a first attempt to export the technology used in electrophysiology laboratories into outpatient settings, we have recently developed a wearable mobile device, termed “μEMG” [10], in the form of a wrist-watch-like support, with two pairs of surface electrodes that record the contraction of the antagonist muscles in the forearm. The recorded data are accessed through a mobile application. This technique closely resembles the process of recording tremors using sEMG. Despite its portability advantage, which allows its use also and in the absence of an electromyograph, the μEMG device still requires the use of surface electrodes for muscle recording, and needs dedicated expertise for the correct application of electrodes on the forearm antagonist muscles and for setting various acquisition parameters (signal-to-noise threshold, duration of the recording, etc.).

The need for simpler and cost-effective approaches in tremor analysis has led to the development of many wearable devices based on inertial sensors, which do not require any specific expertise (differently from EMG-based approaches) and can also be used to evaluate tremors at home or in primary-care centers. Most of the proposed solutions so far have focused on detecting tremor frequency and power, which are of great value for assessing tremor severity. On the other hand, there is a lack of simple devices assessing the phase or pattern of tremors using inertial sensors and signals [11]. 

The objectives of this study were as follows:

To develop a new low-cost miniaturized wearable device, termed “RT-ring”, which can be placed on a finger of the hand with RT to acquire inertial (accelerometric and gyroscopic) data of tremors.

By analyzing the inertial data collected with the RT-Ring experimental prototype, and employing ML algorithms to identify combinations of inertial features that can accurately differentiate between alternating and synchronous tremor patterns. 

This is a proof-of-concept study testing the feasibility of predicting the muscular contraction pattern of rest tremor using inertial data only, in the absence of surface electromyography. The use of Machine Learning (ML) and Artificial Intelligence (AI) has been of great help in the last decades to automate complex tasks and improve performances in various fields, including in industrial [12] and biomedical [13] applications. Rhythmic muscle activation in the forearm can be regarded as the motor source of periodic, tremulous movements of the hand. Therefore, it is reasonable to look for information about different activation patterns of the motor source by analyzing through ML algorithms the movement of the hand as detected by inertial sensors. This approach has never been described so far and would allow an easier evaluation of tremor patterns, even by operators without EMG expertise.

## 2. Materials and Methods

The overall architecture of the proposed hardware and software solution is graphically reported in Figure 2a, while the block diagram of the RT-Ring device is shown in Figure 2b. Acceleration and gyroscopic rate samples were acquired using an inertial measurement unit (IMU) included into the RT-Ring and sent over Bluetooth Low Energy (BLE) communication to a smartphone running a dedicated mobile app. The mobile app preprocesses and stores the data into binary files. Then, these files are sent over the Internet, through a TCP connection to a backend server, where data are processed and classified using pre-trained machine learning (ML) models. The classification response (alternating or synchronous RT pattern) is sent back to the mobile app and the result is shown to the user.

Following are the details of each hardware and software component of the overall system.

### 2.1. Development of the Wearable Device

The Printed Circuit Board (PCB) design of RT-Ring was based on the Nordic Semiconductors’ nRF52840 microcontroller, a multiprotocol Bluetooth 5.3 System-On-Chip (SOC), packed in the ISP-1807 module by Insight SiP. This module integrates an nRF52840 microcontroller together with 32 MHz and 32 KHz external crystal clocks and an RF antenna in an 8.0 × 8.0 × 1.0 mm package. The microcontroller receives inertial data from an ST Microelectronics LSM6DSL 6-axis inertial measurement unit (IMU) through an I^2^C bus. The LSM6DSL sensor exhibits 3 accelerometric axes and 3 gyroscopic axes. The PCB board is also equipped with a low drop-out (LDO) voltage regulator, a battery charge controller, a fuel gauge, and soft button controller coupled with a normally open (NO) single pole—single throw (SPST) button. The top and bottom PCB layouts of the device are shown in Figure 3a and Figure 3b, respectively.

The assembled PCB is sized 1.5 × 1.5 mm (Figure 3c) and is powered by a 65 mAh Lithium-Polymer battery. Both the PCB and the rechargeable battery are housed in a 3D-printed plastic case and a Velcro strap is used to fix the device on the middle finger of the hand affected with tremor, as shown in Figure 3d.

### 2.2. Firmware and Mobile App

The nRF52840 microcontroller is flashed with Nordic’s SoftDevice, version 6.1.1, implementing the Bluetooth Low Energy (BLE) stack, and with a modified bootloader allowing for Over-the-Air (OTA) Device Firmware Update (DFU). The application firmware was developed using the Arduino Integrated Development Environment (IDE). The .hex file produced by the compiler was merged into an OTA-DFU zip package using the nrfutil programming tool.

The application firmware implements a finite state machine and a set of commands used to interact with the device in order to:

Read the device inclination in terms of pitch and roll angles, with an output data rate of 2 samples/s;

Read scaled acceleration and angular velocity data from all sensing axes, with a default output data rate of 100 sample/s, a default acceleration range of ±2 G, and a default angular velocity range of ±500 deg/s;

Configure sensor ranges and output data rate;

Check battery charge status;

Force reboot of the microcontroller in bootloader mode.

Commands and data are exchanged through the UART characteristic of the BLE wireless communication protocol, using a mobile app named e-RT-Ring. The mobile app was developed using the .NET MAUI programming framework and allows the user to control the device and acquire data. The files recorded from the mobile app are then sent over the Internet to a remote server, where a backend application processes the data and sends back responses to the mobile client. The backend microservice was developed in the Python programming language, using the Flask framework.

### 2.3. Data Acquisition and Processing

The RT-Ring device was positioned on the middle finger of the hand with rest tremor of each subject, on the third phalanx in a proximal position, as shown in Figure 3d. The mobile app guided the operator during the inertial data acquisition, ensuring that the hand with tremor was at rest and with a proper inclination before each recording. Surface EMG signals from the extensor carpi radialis and flexor carpi ulnaris muscles were acquired simultaneously to the inertial data to assess the RT pattern in each recording segment. All surface EMG procedures were performed as previously described [14,15]. A total of 200 recording segments (of 10 s each) were recorded from 40 rest tremor patients. Five 10 s recording segments were acquired from each patient.

As rest tremor is mainly expressed in the 4–8 Hz frequency range [1], the signals were bandpass filtered in the 2–10 Hz band using a 4th order, two-pass zero-phase Butterworth filter, in order to encompass all possible tremor oscillations.

Next, a periodicity check was performed on each recording segment to assess the actual presence of tremor information in the collected samples. This check was performed both in time and frequency domain, on the dominant acceleration axis, i.e., on the axis with the highest amplitude. Then, each segment was accepted for processing if there was a magnitude peak in the range 3–8 Hz and the number of zero-crossings was at least half the expected number of zero-crossings of a sine wave at *f_peak_*, with *f_peak_* being the frequency of such magnitude peak. The traces that did not respect these criteria were discarded.

After this quality check, 168 segments were selected for processing. EMG traces corresponding to these inertial recording segments were visually inspected by two independent raters (a neurologist and a technician expert in tremor analysis), and the RT contraction patterns were classified as “A” (alternating) or “S” (synchronous); a third rater evaluated the traces in case of discrepancy. Ninety-one segments were labeled as “A” and 77 segments were labeled as “S” according to visual inspection of RT pattern on corresponding EMG recordings.

In Figure 4, inertial signals acquired from 2 tremulous subjects with different patterns are shown. Figure 4a–d show inertial signals from a subject with alternating tremor pattern (a–b unfiltered and c–d filtered), while signals from a subject with synchronous tremor pattern are shown in Figure 4e–h (e–f unfiltered and g–h filtered).

For each recording segment, extracted inertial data included time and frequency domain features. The IMU included into the RT-ring recorded 6 different signals: *a_x_*, *a_y_*, *a_z_* accelerations, expressed in (g), and *w_x_*, *w_y_*, *w_z_* angular rates, expressed in (deg/s). By considering a typical flexo-extension movement as mainly involving the *y* and *z* accelerometric axes and the *x* gyroscopic axis, the following frequency and time domain features were computed from the *a_y_*, *a_z_*, and *w_x_* signals: 

Tremor frequency from single axes power spectral densities (PSD) and from *a_y_-a_z_* cross power spectral density (cPSD);

Pearson’s correlation coefficient between *a_y_* and *a_z_* axes;

Power spectral amplitudes at tremor frequency, evaluated on a*_y_-a_z_* cPSD and on *w_x_* PSD;

Powers and bandwidths of 3 dB; 

a*_y_-a_z_* magnitude squared coherence;

One-second coherence, defined as the mean of second-by-second ratios between 3 dB power and total power;

Peak-to-peak amplitudes from all axes;

The sum of the first two harmonics on *a_z_* axis;

Detail *d*_3_, *d*_2_, *d*_1_, and approximation *a_3_* coefficients were computed from ‘db2’, 3 levels, wavelet decomposition on each sensing axis.

Power spectral densities and cross-power spectral densities were computed using MATLAB/Octave pwelch and cpsd functions, with 3 s Hanning windowing, 75% overlap, and 2^16^ FFT points. The same parameters were used for the computation of magnitude squared coherence by means of the mscohere function. Wavelet decomposition was performed using the Large Time/Frequency Analysis Toolbox (LTFAT) for MATLAB/Octave.

A total of 63 features were computed for “A” and “S” segments. All the features were compared using either Student’s *t*-test or Wilcoxon rank sum test, after checking for normality using the Shapiro–Wilk test. All *p* values were corrected according to Benjamini–Hochberg method for false discovery rate (FDR). The significance level was set at *p* = 0.05.

### 2.4. Machine Learning Algorithms

We used machine learning (ML) algorithms based on inertial data to estimate the tremor pattern. The visually inspected pattern of RT on EMG traces was used as reference standard, and the performance of two ML algorithms, Random Forest (RF) [16] and Extreme Gradient Boosting (XGB]) [17], in distinguishing recording segments with alternating pattern from those with synchronous pattern, was assessed using combinations of inertial features.

The dataset of 168 recording segments was split into two random, 50% subsets of 84 segments each with a balanced percentage of alternating and synchronous segments. The first subset was used to train the ML models, and the second independent dataset was used as a testing set to calculate the classification performance of the models.

As a first step, inertial features from the training set were sorted in descending order, according to their importance as measured by RF and XGB algorithms. Then, RF and XGB models were trained to perform a 80–20%, 5-fold cross-validation, with hyperparameters tuning. Classification performances of trained models in predicting alternating (A) or synchronous (S) patterns were then assessed on the independent testing set. The tuning of hyperparameters is reported in Table 1.

Processing and feature extraction were performed in GNU Octave and R programming languages. ML models were trained using the R Classification And REgression Training (caret) library [18].

### 2.5. Backend Application

Finally, the best-performing RF and XGB models were selected and embedded in a software tool, designed as a web microservice. This tool was installed on a remote server, to work as an application backend. The backend application was developed in the Python programming language using the Flask framework and implemented a threaded execution to serve multiple incoming requests at the same time.

A block diagram of the whole preprocessing, training, and validation process is shown in Figure 5.

The whole preprocessing and ML training and testing process (Figure 5a) allowed us to train the best-performing RF and XGB models and to evaluate their classification performances. The backend server application (Figure 5b) takes a single segment as input and utilizes one of the best models evaluated in the training phase to obtain a classification response. This response is sent to the mobile application, or a “fail” message is sent if the frequency check is not passed.

## 3. Results

Means and standard deviations and comparisons of computed features are shown in Table 2. 

After FDR correction, 58 out of 63 features were significantly different between the “A” and “S” segments. Before training ML models, two inertial feature sets were defined: **F**_RF_ = {f_RF_^(1)^, f_RF_^(2)^, …, f_RF_^(N)^} and **F**_XGB_ = { f_XGB_^(1)^, f_XGB_^(2)^, …, f_XGB_^(N)^}, including all the initial features sorted by decreasing importance metrics computed by RF and XGB algorithms, respectively. The RF models were trained on all sorted subsets of **F**_RF_, starting with the first feature and then adding one feature at a time. Similarly, XGB models were trained on all sorted subsets of **F**_XGB_.

The model with the highest testing classification accuracy was selected as the best-performing model for RF and XGB, respectively. The best-performing RF model was trained on the first 6 features of **F**_RF_, while the best XGB model was trained on the first 13 features of **F**_XGB_. Optimal feature subsets, together with their computed importance, for RF and XGB models are shown in Figure 6a and Figure 6b, respectively.

Overall accuracy, Cohen’s kappa, sensitivity, specificity, positive predictive value (ppv), and negative predictive value (npv), measured in the training dataset (5-fold cross-validation) and in the independent testing dataset, are reported in Table 3. Performance metrics (accuracy, kappa, sensitivity, and specificity) of all training folds are reported in Table 4.

Cohen’s kappa showed a substantial reliability (kappa > 0.60) of the classifiers when compared to random guessing.

Overall performances of the two models are comparable, even though RF performs slightly better than XGB. Moreover, the RF model used a smaller number of features.

The training Receiver Operating Characteristic (ROC) curves evaluated from the 5-fold 80–20% cross-validation are shown in Figure 7a,b. Figure 7c shows the calibration plots, assessing the agreement between predictions and observations, while the testing ROC curves are shown in Figure 7d.

The mean training Area Under the Curve (AUC) was 0.91 for Random Forest and 0.95 for XGBoost, respectively. On the testing dataset, Random Forest reached an AUC of 0.96, while XGBoost had an AUC of 0.97.

The mean processing time for each segment is less than 5 s, including transfer time to and from the backend server.

## 4. Discussion

In this study, we developed a new compact wearable device for the assessment of rest tremor. This device is termed “RT-Ring” since it can be worn as a ring around a finger of the hand with RT. The main aim of this miniaturized device is to estimate the tremor pattern through machine learning technology using only inertial (accelerometric and gyroscopic) data, in the absence of EMG.

There is great interest in assessing the RT pattern in tremulous disorders. Previous studies demonstrated a strong association between this RT feature and the presence/absence of striatal dopaminergic damage, making the RT pattern of high relevance in the differential diagnosis between Parkinsonian and non-Parkinsonian rest tremor syndromes. The surface EMG is the only approach suitable to evaluate the tremor pattern since it is possible to visualize the temporal shift of contraction burst of antagonistic muscles generating the tremor. Unfortunately, surface EMG requires technology and expertise in tremor analysis, making this approach suitable only for specialists with an interest in tremors. In the last decade, there has been a growing effort to use inertial signals for medical purposes since they can be acquired through MEMS accelerometers and gyroscopes, which can be easily included into wearables or even mobile phones.

The main applications of inertial sensors in tremulous patients described so far include wearable solutions for detecting tremors [19,20,21,22,23,24,25,26,27] or quantitatively assessing tremor severity [28,29,30,31,32,33,34,35,36,37,38,39]. Some devices have been tested in laboratory settings, with the aim of supporting the clinical evaluation of tremors [19,20,21,22,23,24]; and others have been tested in home-based settings with the aim of detecting tremors in the context of daily life [25,26,27], with the potential usefulness of an early detection of tremulous disorders if employed for large-scale population screening. Finally, there has been a growing interest in developing wearable devices to assess tremor severity at home [34,35,36,37,38,39], with obvious significant advantages in monitoring the efficacy of therapies during the whole day, beyond the clinical consultation [28,29,31,34].

A few devices based on inertial data have also been proposed to distinguish among different causes of tremor [40,41,42,43,44,45,46,47,48], mainly between Parkinson’s disease (PD) and essential tremor (ET), but there is still much room for improvement in this field. Most of the proposed devices explored the characteristics of rest tremor in PD and postural tremor in ET [40,41,42,43,44,45,46], which are different by definition, and a few studies focused on postural tremor in both diseases [46,47] or a combination of different tremors [48], while studies focusing on tremor at rest are lacking. In addition, these previous studies mainly explored the potential usefulness of data based on tremor power or frequency [40,41,42,43,46,47] for the differential diagnosis, and none of the described approaches aimed to estimate the tremor pattern using inertial data.

Machine Learning and Artificial Intelligence have gained increasing importance in dealing with classification and estimation problems [13], trying to find relationships between data that are not immediately seen by humans. By fitting mathematical models to data, scientists and clinicians have been able to automatically perform diagnostic tasks with significant accuracy. In the current study, we employed ML algorithms to estimate the RT pattern using combinations of inertial tremor features, achieving excellent classification performances (AUC above 0.90). The RF algorithm found 3 dB power and tremor spectral amplitude on the reference gyroscopic axis (*w_x_*), together with wavelet decomposition coefficients from accelerometric axis *z*, as the most informative features for discriminating between alternating and synchronous tremor patterns. XGBoost confirmed the importance of most of the features identified by RF, adding 3 dB bandwidth and 1 s coherence evaluated on *w_x_*, together with wavelet decomposition coefficients from *w_x_*, *a_y_*, and *a_z_*, thus confirming that such sensing axes are the most important in capturing tremor characteristics using time–frequency information.

The main limitation of this study is the relatively low number of examined recording segments, which makes this project a pilot study. Future validation studies on a larger amount of data are needed to confirm the performance of the RT-Ring system in predicting the RT pattern and further validate the feasibility of this approach.

## 5. Conclusions

The present work is the first attempt, to our knowledge, to characterize muscle behaviors, commonly assessed by electromyographic approaches, using inertial data and combining such data into machine learning models. The excellent classification performances reached by the classifiers, namely, AUC = 0.96, accuracy = 0.92 for RF; and AUC = 0.97, accuracy = 0.89 for XGB, demonstrated that different electromyographic tremor patterns have their counterparts in terms of rhythmic movement features. This may allow the replacement of expensive, complex, and time-consuming electromyographic examinations with a simpler, cheaper, and faster (within a few seconds) evaluation through a tiny wearable device.

We have carried on a full stack development process, from electronic design to the Internet of Things, to devise and test a new wearable device (RT-Ring) and a software system for the automatic characterization of rest tremor. The adoption of a MEMS inertial measurement unit allowed us to design a miniaturized device, which can be easily worn and used by non-specialized operators or even by patients. Future studies will include the comparison of different ML algorithms, using automated tools for model tuning and optimization [12], and the assessment of the RT-ring performance in distinguishing patients with different rest tremor syndromes at the individual level.

## 6. Patents

Italian patent application no. 102021000019793 was filed on 2021/07/26.

## Figures and Tables

**Figure 1 bioengineering-10-01025-f001:**
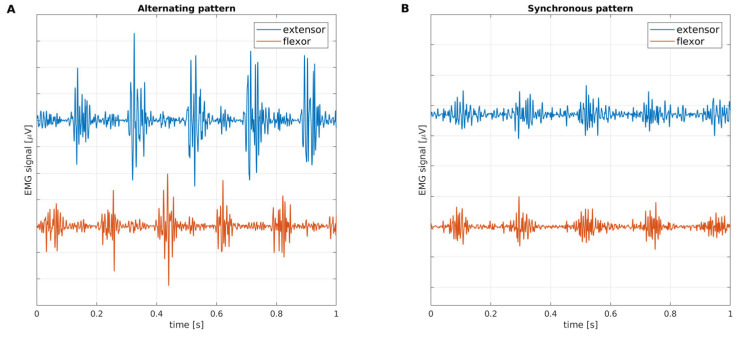
Examples of tremor activation patterns in electromyography (EMG) signals from antagonist muscles of the forearm: (**A**) alternating pattern, when flexor and extensor tremor bursts are phase-shifted; and (**B**) synchronous pattern, when flexor and extensor tremor bursts are in phase. Muscle bursts occur periodically; their repetition frequency is the characteristic frequency of tremor.

**Figure 2 bioengineering-10-01025-f002:**
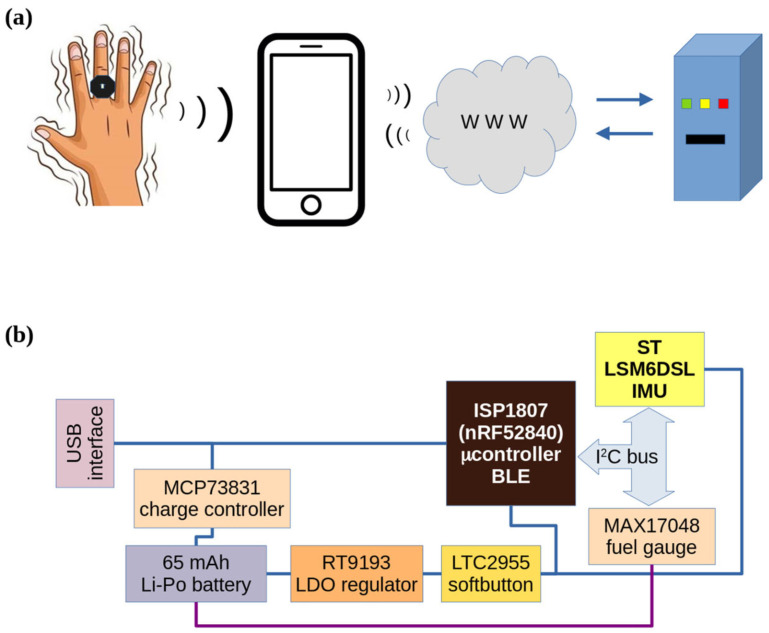
(**a**) Overall signal generation and processing system architecture: signals generated by RT-Ring are acquired by a mobile app and then sent to the backend server through a TCP connection over the mobile network. Processing output is then sent back to the mobile app. (**b**) Block diagram of the RT-Ring wearable device: the core of the device is represented by the ST Microelectronics LM6DSL 6-axis Inertial Measurement Unit (IMU) and by the ISP1807 module, incorporating a nRF52840 microcontroller, with 32 KHz and 32 MHz crystal clocks and BLE antenna. An MCP73831 charge controller provides charging current from USB interface to a 65 mAh Lithium-Polymer batter; an RT9193 Low Dropout regulator converts 3.7 V battery output to 1.8 V and delivers power to all other components. Power delivery is activated by a LTC2955 soft button controller. A MAX17048 digital fuel gauge accurately measures battery charge and communicates with the microcontroller.

**Figure 3 bioengineering-10-01025-f003:**
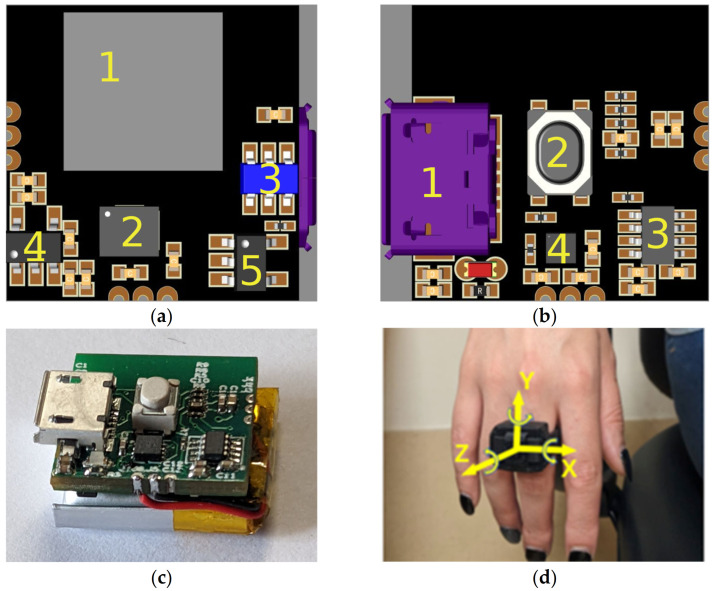
Layout of the PCB: (**a**) Top layer, with (1) ISP-1807 module, (2) ST LSM6DSL IMU, (3) (optional) protection diodes for D0 and D1 USB lines, (4) LDO voltage regulator, and (5) battery charge controller; (**b**) bottom layer, with (1) USB socket, (2) pushbutton, (3) soft-button controller, and (4) fuel gauge; (**c**) assembled PCB with battery; (**d**) device worn on a patient’s middle finger. PCB: Printed Circuit Board; IMU: Inertial Measurement Unit; USB: Universal Serial Bus; LDO: Low Drop-Out.

**Figure 4 bioengineering-10-01025-f004:**
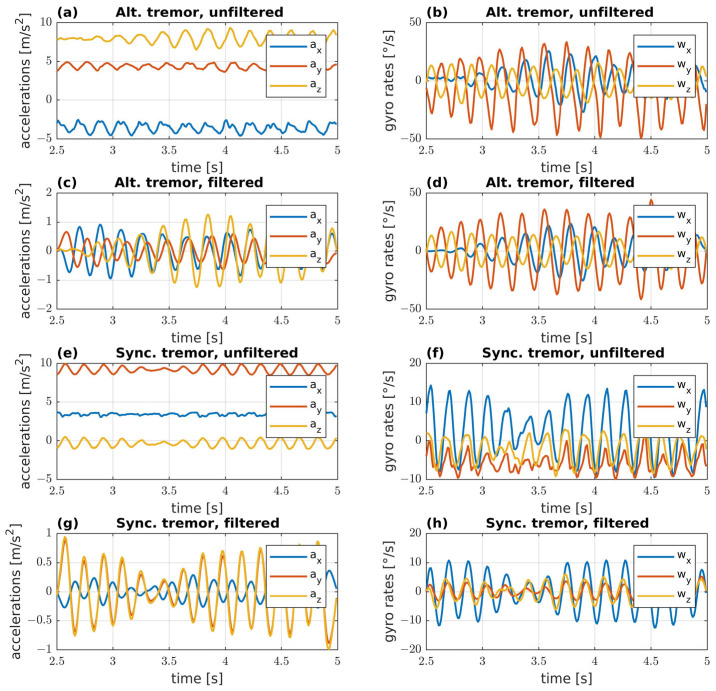
Signals acquired by RT-Ring from two subjects with different tremor activation patterns: (**a**) raw accelerations, (**b**) raw angular velocities, (**c**) filtered accelerations, and (**d**) filtered angular velocities from a subject with alternating tremor pattern; (**e**) raw accelerations, (**f**) raw angular velocities, (**g**) filtered accelerations, and (**h**) filtered angular velocities from a subject with synchronous tremor pattern. Filtered signals have been processed using a 4th order, two-pass, zero-phase Butterworth filter.

**Figure 5 bioengineering-10-01025-f005:**
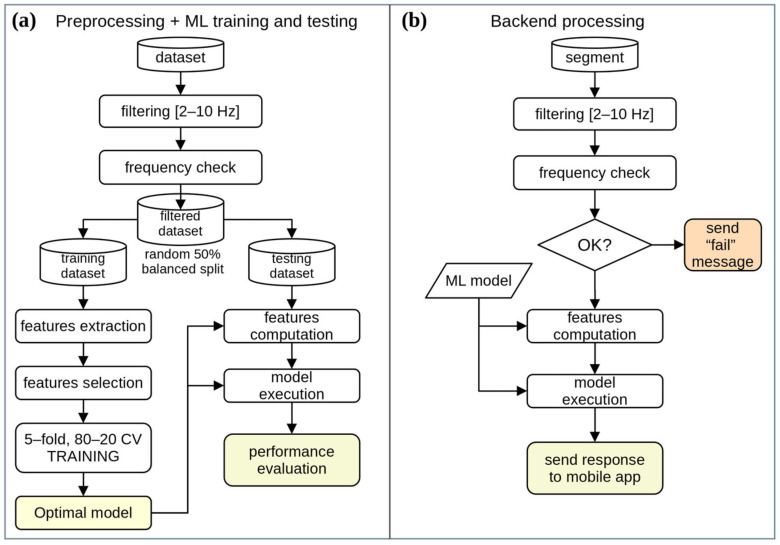
Block diagrams of (**a**) preprocessing and Machine Learning process for the evaluation of optimal models; (**b**) processing of a single segment on the backend server.

**Figure 6 bioengineering-10-01025-f006:**
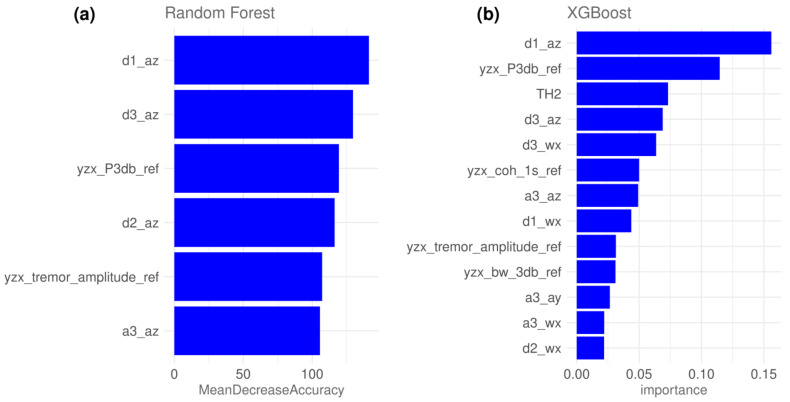
Importance of optimal features subsets used for training Random Forest (**a**) and XGBoost (**b**) best models.

**Figure 7 bioengineering-10-01025-f007:**
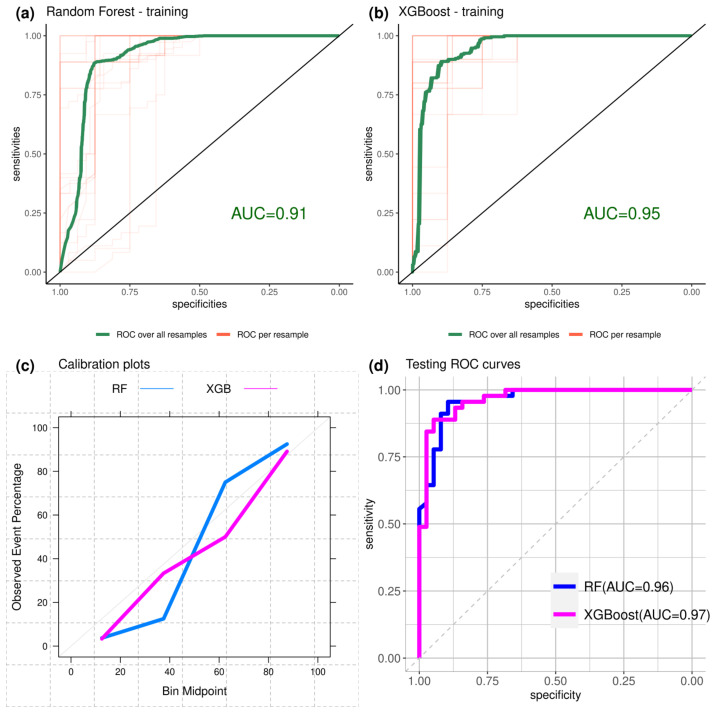
(**a**) ROC curve from Random Forest training with 5-fold 80–20% cross-validation, mean curve is plotted in green color, mean AUC is 0.91; (**b**) ROC curve from XGBoost training with 5-fold 80–20% cross-validation, mean curve is plotted in green color, mean AUC is 0.95; (**c**) calibration plots, assessing the agreement between observations and predictions by the two classifier models; (**d**) ROC curves evaluated on the testing set, showing AUC = 0.96 and AUC = 0.97 for RF and XGB, respectively.

**Table 1 bioengineering-10-01025-t001:** Hyperparameters tuning of Random Forest and eXtreme Gradient Boosting models.

Model	Hyperparameters	Definition
RF	Mtry = 1	Number of variables randomly sampled as candidates at each split
XGBoost	Nrounds = 300 Max.depth = 4 Eta = 0.05 Gamma = 0.0 Col.sample_by_tree = 0.4 Child_weight = 1 Subsample = 0.75	Maximum number of iterations/trees to grow Depth of the tree Learning rate Regularisation (preventing overfitting) Controls the number of features supplied to a tree Leaf threshold for stopping tree splitting Controls the number of samples supplied to a tree

RF: Random Forest; XGBoost: eXtreme Gradient Boosting.

**Table 2 bioengineering-10-01025-t002:** Frequency domain, time domain, and time–frequency domain inertial features used in machine learning models for the classification between Synchronous and Alternating tremor patterns.

Features					
Description		Synchronous Pattern	Alternating Pattern	*p*-Value	
*a_y_-a_z_* cross-spectral frequency (Hz)		5.63 ± 0.69	4.96 ± 0.65	<0.001	^#^
*a_y_-a_z_* cross-spectral amplitude (m^2^/s^4^/Hz)		0.38 ± 1.03	10.70 ± 18.32	<0.001	^#^
*w_x_* spectral amplitude ((°/s)^2^/Hz)		338 ± 1619	13656 ± 23240	<0.001	^#^
*a_y_-a_z_* magnitude squared coherence		0.81 ± 0.27	0.96 ± 0.09	<0.001	^#^
*a_y_-a_z_* axes correlation		0.55 ± 0.64	0.93 ± 0.15	<0.001	^#^
*a_y_-a_z_* phase difference (°)		45.35 ± 54.50	11.84 ± 17.36	<0.001	^#^
3 dB power, *a_y_-a_z_* cross-spectrum (m^2^/s^4^)		0.18 ± 0.49	4.80 ± 8.34	<0.001	^#^
3 dB powers (m^2^/s^4^)	*w_x_*	156 ± 693	6144 ± 10628	<0.001	^#^
	*a_y_*	0.26 ± 0.91	4.95 ± 9.04	<0.001	^#^
	*a_z_*	0.14 ± 0.30	6.18 ± 11.81	<0.001	^#^
3 dB bandwidth, *a_y_-a_z_* cross-spectrum (Hz)		0.65 ± 0.24	0.57 ± 0.09	0.016	^#^
3 dB bandwidths (Hz)	*w_x_*	0.74 ± 0.33	0.58 ± 0.14	0.001	^#^
	*a_y_*	0.72 ± 0.29	0.59 ± 0.14	0.001	^#^
	*a_z_*	0.68 ± 0.26	0.57 ± 0.08	0.003	^#^
*y,z* spectral amplitudes (m^2^/s^4^/Hz)	*a_y_*	0.39 ± 0.73	2.46 ± 2.45	<0.001	^#^
	*a_z_*	0.36 ± 0.52	2.88 ± 2.83	<0.001	^#^
Mean 1 s coherence, *a_y_-a_z_* cross spectrum		0.71 ± 0.07	0.75 ± 0.04	<0.001	^#^
Mean 1 s coherence, *w_x_* spectrum		0.72 ± 0.05	0.76 ± 0.02	<0.001	^#^
Var. 1 s coherence, *y-z* cross spectrum		0.04 ± 0.05	0.01 ± 0.02	<0.001	^#^
Var. 1 s coherence, *w_x_* spectrum		0.04 ± 0.03	0.01 ± 0.02	<0.001	^#^
Skew. 1 s coherence, *a_y_-a_z_* cross spectrum		−0.47 ± 0.74	−0.37 ± 0.72	0.582	^#^
Skew. 1 s coherence, *w_x_* spectrum		−0.38 ± 0.72	−0.32 ± 0.71	0.631	*
Kurt. 1 s coherence, *a_y_-a_z_* cross spectrum		2.46 ± 1.00	2.46 ± 1.01	0.962	^#^
Kurt. 1 s coherence, *w_x_* spectrum		2.48 ± 1.06	2.40 ± 0.80	0.881	^#^
Single-axis frequencies (Hz)	*a_x_*	5.52 ± 0.84	4.97 ± 0.73	<0.001	^#^
	*a_y_*	5.63 ± 0.71	4.96 ± 0.65	<0.001	^#^
	*a_z_*	5.61 ± 0.65	4.95 ± 0.65	<0.001	^#^
	*w_x_*	5.57 ± 0.78	4.91 ± 0.63	<0.001	^#^
	*w_y_*	5.51 ± 0.86	4.97 ± 0.81	<0.001	^#^
	*w_z_*	5.63 ± 0.66	4.98 ± 0.73	<0.001	^#^
Single-axis accelerometer peak-to-peak amplitudes (mG)	*a_x_*	93.38 ± 149.64	355.73 ± 349.61	<0.001	^#^
	*a_y_*	85.64 ± 132.39	521.64 ± 517.66	<0.001	^#^
	*a_z_*	77.69 ± 90.18	594.47 ± 562.86	<0.001	^#^
Vectorial magnitude of peak-to-peak accelerometer amplitudes (mG)		157.76 ± 212.64	919.24 ± 782.95	<0.001	^#^
Single-axis gyroscope peak-to-peak amplitudes (°/s)	*w_x_*	17.54 ± 32.72	180.59 ± 174.16	<0.001	^#^
	*w_y_*	26.43 ± 56.13	76.24 ± 81.57	<0.001	^#^
	*w_z_*	31.97 ± 66.05	84.77 ± 82.09	<0.001	^#^
Vectorial magnitude of peak-to-peak gyroscope amplitudes (°/s)		49.07 ± 90.55	224.69 ± 196.97	<0.001	^#^
Sum of first 2 harmonics on axis *a_z_* (m^2^/s^4^/Hz)		0.12 ± 0.26	5.66 ± 10.58	<0.001	^#^
*a_x_*, wavelet approx. coefficient (m/s^2^)	a_3_	7.17 ± 3.36	9.61 ± 5.08	0.004	^#^
*a_x_*, wavelet detail coefficients (m/s^2^)	d_3_	0.62 ± 0.82	2.39 ± 2.04	<0.001	^#^
	d_2_	0.24 ± 0.24	1.04 ± 1.02	<0.001	^#^
	d_1_	0.10 ± 0.09	0.46 ± 0.57	<0.001	^#^
*a_y_*, wavelet approx. coefficient (m/s^2^)	a_3_	24.54 ± 2.62	22.90 ± 5.54	0.158	^#^
*a_y_*, wavelet detail coefficients (m/s^2^)	d_3_	0.83 ± 0.79	3.11 ± 2.44	<0.001	^#^
	d_2_	0.48 ± 0.27	1.10 ± 0.96	<0.001	^#^
	d_1_	0.26 ± 0.13	0.45 ± 0.42	0.001	^#^
*a_z_*, wavelet approx. coefficient (m/s^2^)	a_3_	7.92 ± 5.13	12.83 ± 6.32	<0.001	^#^
*a_z_*, wavelet detail coefficients (m/s^2^)	d_3_	0.58 ± 0.55	3.55 ± 2.91	<0.001	^#^
	d_2_	0.22 ± 0.16	1.27 ± 1.27	<0.001	^#^
	d_1_	0.10 ± 0.07	0.52 ± 0.75	<0.001	^#^
*w_x_*, wavelet approx. coefficient (°/s)	a_3_	19.61 ± 30.67	162.47 ± 149.40	<0.001	^#^
*w_x_*, wavelet detail coefficients (°/s)	d_3_	12.11 ± 19.61	100.66 ± 92.52	<0.001	^#^
	d_2_	3.65 ± 6.04	29.83 ± 33.77	<0.001	^#^
	d_1_	1.16 ± 2.34	8.61 ± 12.26	<0.001	^#^
*w_y_*, wavelet approx. coefficient (°/s)	a_3_	31.61 ± 40.73	76.36 ± 77.29	<0.001	^#^
*w_y_*, wavelet detail coefficients (°/s)	d_3_	17.28 ± 33.33	52.65 ± 51.71	<0.001	^#^
	d_2_	5.14 ± 8.72	20.67 ± 23.12	<0.001	^#^
	d_1_	1.32 ± 2.13	7.20 ± 9.89	<0.001	^#^
*w_z_*, wavelet approx. coefficient (°/s)	a_3_	28.76 ± 48.37	80.76 ± 75.45	<0.001	^#^
*w_z_*, wavelet detail coefficients (°/s)	d_3_	20.86 ± 41.14	51.00 ± 44.17	<0.001	^#^
	d_2_	5.55 ± 10.08	15.75 ± 13.79	<0.001	^#^
	d_1_	1.30 ± 2.30	4.73 ± 5.17	<0.001	^#^

^#^ Wilcoxon rank sum test, * Student’s *t*-test, *p* values adjusted according to Benjamini–Hochberg FDR correction method.

**Table 3 bioengineering-10-01025-t003:** Classification performances in training and testing datasets for RF and XGB models.

Model	Metric	Training	Testing
Random Forest	accuracy	0.88 (0.87–0.90)	0.92 (0.83–0.97)
	kappa	0.76	0.83
	sensitivity	0.89	0.96
	specificity	0.88	0.87
	ppv	0.89	0.90
	npv	0.87	0.94
XGBoost	accuracy	0.89 (0.86–0.92)	0.89 (0.80–0.95)
	kappa	0.79	0.78
	sensitivity	0.89	0.96
	specificity	0.90	0.82
	ppv	0.91	0.86
	npv	0.88	0.94

**Table 4 bioengineering-10-01025-t004:** Performance metrics of ML models in all training folds.

Model	Fold	Accuracy	Kappa	Sensitivity	Specificity
Random forest	1	0.94	0.88	0.90	1.00
	2	0.82	0.65	0.78	0.88
	3	0.87	0.73	1.00	0.72
	4	0.91	0.82	1.00	0.81
	5	0.87	0.74	0.78	0.97
	6	0.91	0.82	0.89	0.94
	7	0.85	0.69	0.90	0.79
	8	0.71	0.41	0.67	0.75
	9	0.97	0.94	1.00	0.94
	10	0.91	0.82	1.00	0.81
	11	0.82	0.64	0.89	0.75
	12	0.97	0.94	1.00	0.94
	13	0.91	0.82	0.85	1.00
	14	0.76	0.53	0.67	0.88
	15	0.94	0.88	1.00	0.88
	16	0.91	0.82	0.85	1.00
	17	0.76	0.52	0.89	0.63
	18	0.90	0.79	0.92	0.88
	19	1.00	1.00	1.00	1.00
	20	0.84	0.68	0.78	0.91
	21	0.84	0.67	0.83	0.86
	22	0.85	0.71	0.83	0.88
	23	0.94	0.88	1.00	0.88
	24	0.94	0.88	0.89	1.00
	25	0.88	0.76	0.89	0.88
	mean	**0.88**	**0.76**	**0.89**	**0.88**
	std. dev.	**0.07**	**0.14**	**0.10**	**0.10**
XGBoost	1	0.91	0.82	0.97	0.84
	2	0.90	0.79	0.86	0.94
	3	0.90	0.79	0.87	0.93
	4	0.88	0.76	0.87	0.90
	5	0.90	0.80	0.84	0.97
	6	0.96	0.91	0.95	0.97
	7	0.88	0.76	0.86	0.91
	8	0.90	0.79	0.89	0.90
	9	0.90	0.79	0.89	0.90
	10	0.90	0.79	0.89	0.90
	11	0.91	0.82	0.89	0.94
	12	0.90	0.79	0.86	0.94
	13	0.87	0.73	0.86	0.88
	14	0.84	0.67	0.84	0.84
	15	0.87	0.73	0.86	0.87
	16	0.88	0.76	0.86	0.90
	17	0.90	0.79	0.86	0.94
	18	0.90	0.79	0.89	0.91
	19	0.90	0.79	0.92	0.87
	20	0.90	0.79	0.92	0.87
	21	0.93	0.85	0.95	0.90
	22	0.90	0.79	0.89	0.90
	23	0.90	0.80	0.89	0.92
	24	0.88	0.76	0.89	0.88
	25	0.76	0.53	0.78	0.75
	mean	**0.89**	**0.78**	**0.88**	**0.90**
	std. dev.	**0.03**	**0.07**	**0.04**	**0.05**

## Data Availability

The data presented in this study are available on request from the corresponding author. The data are not publicly available due to privacy reasons.

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
