# Peer review of "Development of a New Wearable Device for the Characterization of Hand Tremor"

_bioengineering, 2023, doi:10.3390/bioengineering10091025_

Round 1

Reviewer 1 Report

Vescio et al. developed a wearable device, called "RT-Ring,"  for differentiating between the alternating and synchronous rest tremor (RT) patterns using only inertial data. The miniaturized device incorporates a 6-axis inertial measurement unit and a Bluetooth Low Energy microprocessor. And the prediction of the tremor pattern is carried out on a remote server using machine learning (ML) models. The authors validated two decision tree-based algorithms, namely XGBoost and Random Forest, which were trained on features extracted from the inertial data. These algorithms achieved a classification accuracy of 92% and 89%, respectively, showing potentially assisting diagnoses of Parkinson’s disease (PD) and Essential Tremor (ET).

I suggest accepting the paper after some revisions that do not significantly affect the science of the paper.

1. The authors can perform clinical experiments aimed at measuring the rest tremor in different patients, encompassing those with Parkinson's disease and Essential Tremor. These experiments will enable them to accurately calculate the true positive, true negative, and prediction sensitivity of the device.

Again, the science is solid, this is a nice practical advancement.

Author Response

We thank the Reviewer for these valuable comments and suggestions. The aim of the current study was to develop a new method for estimating tremor pattern using inertial data only, but they anticipated what we are actually working on. We are conducting new experiments in order to investigate the performance of the RT-Ring on a large cohort of patients with Parkinson’s Disease (PD) and patients with Essential Tremor with rest temor (rET). The aim of this new study will be the validation of the classification performances of this device in distinguishing PD from ET on an individual basis. We included a sentence on this point in the manuscript.

Reviewer 2 Report

1.  The abstract has to highlight the contribution and novelty clearly.

2. The keywords has to be unique and relevant.

3. The literature review is weak and not streamlined. The literature has to be focused on previous devices that have been used in this sense.

4. The authors have to explain the phrase "None of the existing devices, how-74 ever, …… [10]". Why the reference has been added at the end of this phrase.

5. The contribution has to be states clearly in points at the end of Introduction part. In addition, please, this part has to be rewritten in more compact form.

6. Figure 1 has to be supported by block diagram. In addition, the figure has to be supported by illustration (on the figure itself).

7. The authors have used many tools like smart phone, and the device located at the hand of patients. I think, they could use software supporting smart phone to discard the role of PC. In other words, some machine learning-based processes have to be built-in at the smartphone. 

8.  The article lacks the mathematical analysis.

9.  Figure 4 has low resolution and weak explanation.

10. The authors has to conduct a comparison study with other technology.

11. The filtering of data acquisition has not addressed. The presence of accelerometers is the source of this problem.

12. The authors have to conduct comparison with other classification techniques in the literature. Otherwise, they have to add future work to suggest them. I suggest to refer to the classifier used in   

doi.org/10.3390/pr11051507

13. The conclusion is descriptive. It is void of quantitative and numerical improvement and comparison.

Minor editing of English language required.

Author Response

We thank the Reviewer for their valuable comments and suggestions, that helped us improve the quality of our manuscript. Following are the changes we have made to the manuscript, according to the Reviewer’s precious suggestions.

  1. The abstract has to highlight the contribution and novelty clearly.
    We have added a few sentences to the abstract, in order to highlight the novelty of our study.

  1. The keywords has to be unique and relevant.
    We have rewritten pertinent keywords that are specific to the article and reasonably common within the subject discipline. These are: tremor pattern; inertial signals; wearable device; machine learning; pattern prediction.

  2. The literature review is weak and not streamlined. The literature has to be focused on previous devices that have been used in this sense.
    We have significantly expanded the literature review in the discussion paragraph and associated references, focusing on similar devices and solutions for the assessment of tremor.

  1. The authors have to explain the phrase "None of the existing devices, how-74 ever, …… [10]". Why the reference has been added at the end of this phrase.
    We rephrased this sentence to make it clear. The reference is a recent review work conducted by our research group on the available devices for the evaluation of tremor.

  2. The contribution has to be states clearly in points at the end of Introduction part. In addition, please, this part has to be rewritten in more compact form.
    The contribution of this paper has been clearly stated in points at the end of the introduction section. Changes have been made to this section in order to make it more compact and readable.

  3. Figure 1 has to be supported by block diagram. In addition, the figure has to be supported by illustration (on the figure itself).
    Figure 1 has been split into 2 subfigures (Figures 1A and 1B) and a block diagram has been added, showing the architecture of the wearable device (Figure 1b). Explanations have been added to the caption, making it more readable. A new figure, named Figure 3, has been added, showing the block diagram of the preprocessing and machine learning process. Old Figure 3 and Figure 4 have been renamed Figure 4 and Figure 5, respectively.

  4. The authors have used many tools like smart phone, and the device located at the hand of patients. I think, they could use software supporting smart phone to discard the role of PC. In other words, some machine learning-based processes have to be built-in at the smartphone.
    This is a very interesting argument. For practical reasons, we have chosen to deploy preprocessing tasks and ML computations on a remote machine running a Flask microservice. This choice allowed for a quicker development phase. Moreover, this architecture helps us performing maintenance quicker, without affecting other components of the whole system. Indeed, mobile phones are becoming more and more computationally efficient, but deployment of ML algorithms directly to the Android/iOS environment is still difficult: there are a few frameworks that work well on some development platforms, like TensorFlow Lite, but do not incorporate tree models and other ML models. On .NET development platforms there are a few already cooked recipes, but developing a custom model is still difficult. However, we are planning to move ML to mobile environments in the future, as the technology gets mature and reliable.

  5. The article lacks the mathematical analysis.
    Mathematical details have been added to the Methods section about features computation and statistical analysis. A table has been added to the Supplementary section, showing means, standard deviations and statistical comparisons of computed features among segments with different tremor patterns.

  6. Figure 4 has low resolution and weak explanation.
    Figure 4 has been renamed as Figure 5 and has been replaced by its 600-dpi version, generated by an R script, using ggplot2 library. Figure caption has been expanded with more detailed explanations. Moreover, we corrected a mistyping error in the script for the generation of the plots and updated them.

  7. The authors has to conduct a comparison study with other technology.
    The only available technology for the evaluation of tremor muscular activation pattern is electromyography (EMG), that we use as reference for the machine learning process. No other approaches have been proposed so far for the automated estimation of tremor pattern using inertial signals. Our approach is new and original, and may potentially replace EMG in all those clinical settings where EMG expertise and equipment is not available, allowing for screening on a large scale and simplifying diagnostic procedures.

  8. The filtering of data acquisition has not addressed. The presence of accelerometers is the source of this problem.
    Filtering was performed in the 2-10 Hz band. Description has been added to the Methods section.

  9. The authors have to conduct comparison with other classification techniques in the literature. Otherwise, they have to add future work to suggest them. I suggest to refer to the classifier used in doi.org/10.3390/pr11051507
    This is a very interesting point. We are planning future work and a focused paper to make a comprehensive comparison and assessment of classification models on a larger dataset. We included a sentence on this point in the manuscript.

  10. The conclusion is descriptive. It is void of quantitative and numerical improvement and comparison.
    We have added quantitative results to the conclusion, summarizing the key novelty and outcomes of this work, and projecting to future work and developments.

Round 2

Reviewer 1 Report

The authors addressed all the reviewers' concerns, making substantial changes to improve the overall quality of the work. It can be accepted for publication.

Author Response

We thank the Reviewer for having read and appreciated our work and for his/her valuable comments and suggestions.

Reviewer 2 Report

1.  The authors have to present step-by-step answers to all previous comments. Also, to highlight these changes. It is not clear to use the template of manuscript to address the reviewer's response.

2. This study lacks analysis background. There is no any mathematical basis to introduce the proposed device.

3. The authors have mainly focused on the hardware setup.

4. Figure 1 shows the hardware architecture and there is no idea about the scientific basis or theoretic background behind tremor source or tremor suppression.

5. The authors have never displayed any data or displayed the signals through the pre-processing operations.

6. The training process lacks the behaviours of cost functions.

7. The authors have to conduct comparison with other classification techniques in the literature.

Dear Editor,

I have read the revised version of article and I have reached to the following conclusions.

1. This study is void from any theoretical basis and the authors have focused on the hardware setup as if they are conducting a project; not an academic research.

2. The authors have not displayed the various signals at the preprocessing level. They have presented results without real signals.

3. The authors have to presents step-by-step answers to all comments. The have used the style of editing on the same revised version and this will arise some type of confusion.

As such, I have recommend major corrections to be strictly made; otherwise my decision is the rejection of article.

Thank you for your trust…with best regards

Prof. Dr. Amjad J. Humaidi

Author Response

The complete, point-by-point reply is in the attachment.
